# TAS-GNN: Topology-Aware Spiking Graph Neural Networks for Graph Classification

## Abstract

The recent integration of spiking neurons into graph neural networks has been gaining much attraction due to its superior energy efficiency. Especially because the irregular connection among graph nodes fits the nature of the spiking neural networks, spiking graph neural networks are considered strong alternatives to vanilla graph neural networks. However, there is still a large performance gap for graph tasks between the spiking neural networks and artificial neural networks. The gaps are especially large when they are adapted to graph classification tasks, where none of the nodes in the testset graphs are connected to the training set graphs. We diagnose the problem as the existence of neurons under starvation, caused by the irregular connections among the nodes and the neurons. To alleviate the problem, we propose TAS-GNN. Based on a set of observations on spiking neurons on graph classification tasks, we devise several techniques to utilize more neurons to deliver meaningful information to the connected neurons. Experiments on diverse datasets show up to 27.20% improvement, demonstrating the effectiveness of the TAS-GNN.

## 1 Introduction

Graph neural networks (GNNs) are types of popular neural networks to learn the representations from graphs, which comprise multiple nodes and edges between them. Because of their flexibility to model any kind of connection existing in nature, it has various applications ranging from drug discovery [6, 47, 9], social influence prediction [39, 2], traffic forecasting [3, 7], and recommendation systems [38, 15, 61]. One known challenge of GNNs is their sparse memory and computational pattern. Because many messages are passed between randomly connected nodes, there is a significant inefficiency in processing them with conventional systems [53, 58, 57, 19].

To address the inefficiency, spiking neural networks (SNNs) are considered strong alternatives. Inspired by the way biological behavior of brains, SNNs process information by communicating binary spikes between the neurons. Because SNNs utilize intermittently occurring spikes, they have superior energy efficiency, especially for the domain of GNNs [1].

Although the spiking graph neural network (SGNN) has been recently studied by many researchers [32, 64, 48], we find that its performance experiences a huge drop when adapted to graph classification, compared to that of the conventional GNNs implemented with artificial neural networks (ANNs). Upon closer analysis of the performance degradation, we identify spike frequency deviation of the neurons within the model. In our investigation, many neurons experience *starvation*, which do not emit any spike during the inference. This leads to severe information loss, due to being unable to deliver signals to the subsequent neurons.

Submitted to 38th Conference on Neural Information Processing Systems (NeurIPS 2024). Do not distribute.

Such a problem was less exposed in previous spiking GNNs. This is because the testset nodes are available during the training time (transductive learning [27]) or they are part of the training graph (inductive learning [21]). In such settings, the model could be trained to mitigate the performance drop. However, in graph classification tasks, the graphs are independent of each other, and the testset comprises multiple unseen graphs, aggravating the problem.

Fortunately, our further analysis reveals that such phenomena are related to the topology of the input graphs. We discover that a strong pattern exists among the neurons in the GNN, where 1) neurons in a node have similar behaviors, 2) each feature causes different behaviors, and 3) neurons in high-degree nodes tend to emit more spikes.

Motivated by the observations, we propose to group the neurons according to the degree of the node (*topology-aware group-adaptive neurons*). The neurons in each group adapt the threshold together to steer the firing rate toward ideal rates. To further mitigate the initial value sensitivity problem, we further propose to learn the initial values.

We evaluate TAS-GNN over multiple GNN models and datasets. Experiments reveal that the proposed TAS-GNN achieves superior performance over the baselines, setting a new state-of-the-art method for graph classification. Our contributions are summarized as the following:

- We identify starvation problem of spiking neurons in GNNs for graph classification tasks.
- We observe the spike frequency patterns have a strong correlation with the graph topology.
- Based on the observations, we propose topology-aware group-adaptive neurons, which dynamically adjusts the threshold together with the other neurons in the group to address the spike frequency deviations.
- We propose techniques to reduce the initial value sensitivity caused by the topology-aware group-adaptive neurons.
- We evaluate TAS-GNN on several public datasets and achieve superior performance over existing techniques.

## 2 Background

### 2.1 Spiking Neural Networks and Spike Training

Spiking neural networks (SNNs) are third-generation neural network designs that mimic the human biological neural systems [35]. They use spike-based communication and adopt event-driven characteristics that promote better energy efficiency than current ANNs. Similar to human neural systems, SNNs consist of spiking neurons that can model spatio-temporal dynamics of the actual biological neurons. The early forms of such neuron models are Hodgkin-Huxley neurons [23], which accurately model the biophysical characteristics of the membrane through differential equations. However, its mathematical complexity prohibits its practical use and scalability. Instead, Leaky Integrated-and-Fire (LIF) model finds a middle ground between mathematical simplificity and biological plausibility, and is popularly adopted as the baseline architecture [23]. In the LIF neuron, the weighted sum of input spikes is accumulated over time within the neuron as membrane potential, and the output spike is generated only when the membrane potential exceeds a present threshold value. This is represented as a differential function:

$$\tau \frac{dV(t)}{dt} = -V(t) + I(t), \tag{1}$$

where $V(t)$ denotes the membrane potential value at time $t$, $\tau$ a time constant of membrane, and $I(t)$ is the input from connected synapses at time $t$. To make this time-varying function computationally feasible, we discretize and rewrite it iteratively for sequential simulation as follows:

$$V(t) = V(t-1) + \beta(WX(t) - (V(t-1) - V_{reset})), \tag{2}$$

$$V(t) = V(t)(1 - S(t)) + V_{reset}S(t), \tag{3}$$

$$S(t) = \begin{cases} 1, & \text{if } V(t) \geq V_{th} \\ 0, & \text{otherwise,} \end{cases} \tag{4}$$

where $\beta$ is simplified decay rate constant, $V_{reset}$ is the reset value and $V_{th}$ the threshold for the membrane potential. Note that I(t) is simplified as weighted input WX(t) which can be obtained

through any operations with learnable weights including convolutional operation, self-attention, or a simple MLP. We will denote this process of forwarding through LIF neuron as $SNN(\cdot)$ in this paper.

**Direct SNN Training.** The initial adoption of SNNs was through ANN-SNN conversion, primarily due to their remarkable potential for reducing energy consumption. Various studies have aimed to address the accuracy degradation that occurs during the conversion from ANNS to SNNs [22, 41, 24, 42].

The spike generation by the step function in Equation (4) interfered with direct training without modifying the functions. To bypass the step function, which is non-differentiable and thus unsuitable for backpropagation, several approaches have been proposed [43, 5, 13, 14, 8, 51, 10]. Recent research has demonstrated that directly training SNNs can yield competitive results by addressing the challenges posed by non-differentiability. Our work focuses on directly training graph neural networks (GNNs) with SNNs and exploring a different domain, such as ANN-SNN conversion methods, which do not focus on using backpropagation concepts directly.

## 2.2 Graph Neural Networks

Graph neural networks (GNNs) take graph-represented data as input, which consist of nodes and their connected edges $\mathcal{G} = (V, E)$, with node features $\mathbf{X} \in \mathbb{R}^{|V| \times F}$ and optionally edge features $\mathbf{E} \in \mathbb{R}^{|E| \times D}$. The common GNN architectures follow a message passing paradigm [20], which learns node or edge representations through aggregating information from its neighboring nodes and updating the node features iteratively. Thus a single forward of message passing layer consists of message passing, aggregation, and update: $h_i^{(l+1)} = \phi(h_i^{(l)}, \bigoplus_{j \in \mathcal{N}(i)} \psi(h_i^{(l)}, h_j^{(l)}, e_{ij}))$, where $l$ and $i$ are indices for layer and node, respectively, and $\psi(\cdot)$ denote message passing function. After aggregation of neighboring features, $\phi(\cdot)$ is used for feature update. For graph convolutional network [27], the overall process can be simplified as:

$$X^{(l+1)} = AX^{(l)}W^{(l)}, \tag{5}$$

where the feature matrix is a concatenation of node features $X^{(l)} = [h_0^{(l)} || h_1^{(l)} || ... || h_{(|V|-1)}^{(l)}]^T$ which is updated through iterations of aggregation ($AX$) and combination ($XW$). After iterative updates of $X$ through the layers, the learned node or edge embeddings are passed through additional classification layer for node-level or edge-level predictions.

**Graph Classification** In this paper we put emphasis on graph-level classification tasks where each graph is considered an individual input. Graph classification follows the same node-wise message passing framework to obtain node embeddings, but appends a readout layer to turn them into a single graph embedding:

$$h_G = R(h_i^{(L)} | V_i \in \mathcal{G}), \tag{6}$$

where $R$ denotes readout function. Readout function reduces the node dimension to a single channel regardless of the input size. This is due to the inductive nature of graph classification task where the number of nodes is not known in advance. While all the other GNN layers focus on aggregating only the local features, the readout layer considers the entire graph to generate global features, and is unique to the graph classification tasks. The obtained graph embedding is passed through a classification layer for graph predictions. Graph classification tasks usually hold more difficulty than node-level classification due to its inductive nature, where inference is done on unseen graphs and thus cannot utilize any graph-specific statistics from the train set.

## 2.3 Spiking Graph Neural Networks

In this paper, we adopt conventional SNN designs where LIF neurons are connected through learnabled weights, and apply is to GNN framework [64]. As mentioned in Section 2.2, each GNN layer outputs updated feature matrix $X^{(l+1)} \in \mathbb{R}^{|V| \times F}$. This is converted to spike representation through SNN layer:

$$X^{(l+1)} = SNN(AX^{(l)}W^{(l)}). \tag{7}$$

After passing the GNN layer, all of the updated $h_i^{(l)}$ directly pass through the SNN layer, consist the feature matrix $X^{(l)}$ always contains spike information consistently.

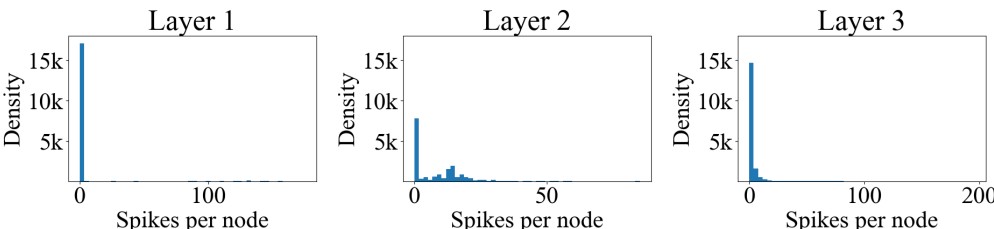

(a) Histogram plotting distribution of total spikes counted over time for each node. X-axis denotes spike counts from each node, while y-axis denotes density of each bin.

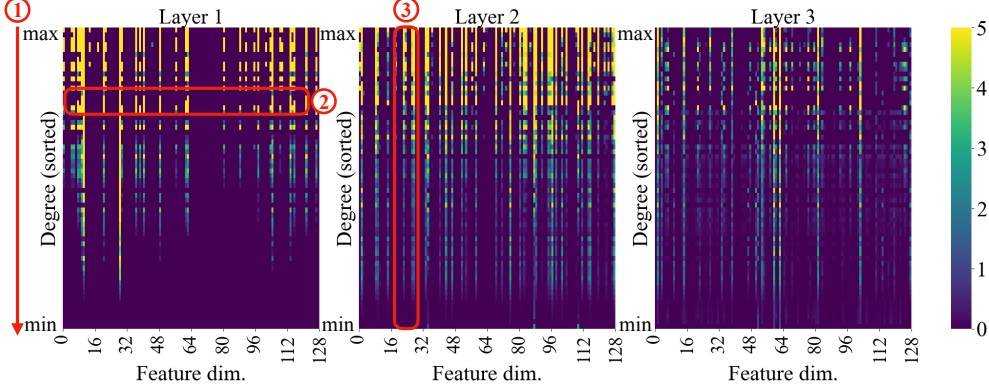

(b) Spike frequency visualization using each layer output. X-axis denotes feature dimension, while y-axis denotes nodes grouped and sorted by degree in descending order, top to bottom. Brighter spots denote higher frequency.

Figure 1: Analysis on spike frequency variation of GCN using IMDB-BINARY [54] dataset.

## 3 Analysis on Spike Frequency Variation of GNNs

To analyze the cause of the accuracy drop, we plot the behavior of the neurons during inference in Figure 1a, on a IMDB-BINARY dataset over five timesteps ($T = 5$). We create a histogram of spike counts created from each node, which is associated with 128 neurons. As depicted in the plot, it is clear that most of the neurons are under starvation. This is caused by the inputs of those neurons being insufficient to reach the threshold, and this leads to severe information loss between the layers. While unveiling the exact dynamics would require more research, we hypothesize that this is caused by the topology of the real-world graphs.

To validate the hypothesis and further investigate the phenomena, we display the spike frequency heatmap of the neurons sorted by the degree of the nodes in Figure 1b. From the heatmap, we make three observations:

① **(Brighter on the top and darker at the bottom)** *High-degree nodes tend to exhibit higher spike frequencies.*

② **(The horizontal strips)** *The spike frequencies are associated with the corresponding nodes.*

③ **(The vertical strips)** *The feature neurons within a node behave differently according to their positions.*

We believe such patterns come from the connectivity of the nodes, and the distinct role of the neurons assigned to each node. The connectivity will affect the number of receiving spikes of neurons associated with each node. It is known that most of the real-world graphs exhibit an extremely skewed distribution of degrees (i.e., power-law distribution [30]). Due to such a characteristic, there are a few nodes with very high degrees, while a majority of nodes have low degrees. Because a GNN layer communicates signals between the neighbors, a high-degree node will likely receive a lot of spikes, while a low-degree node will receive only a few.

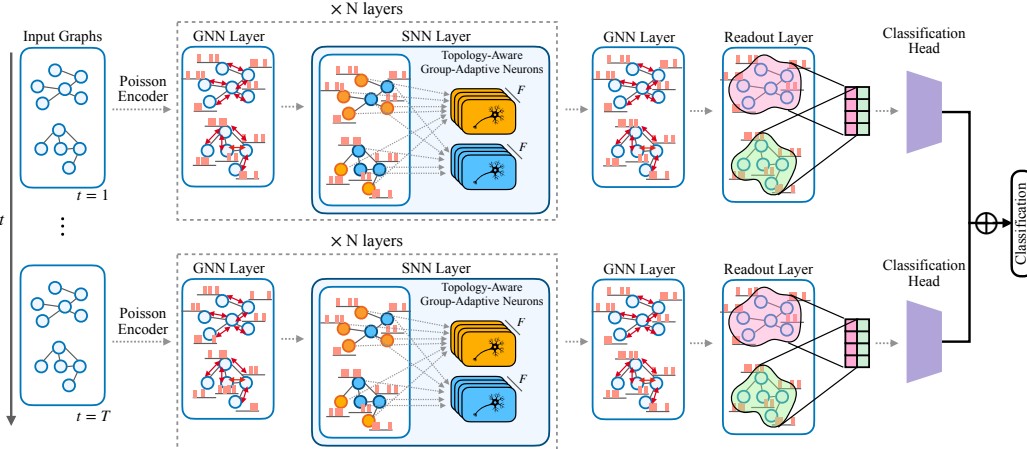

Figure 2: Overall graph classification architecture with proposed methods.

In addition, the neurons assigned to each node are known to have different semantic functionality according to their positions, analogous to *channels* in convolutional neural networks or *heads* in large language models. For example, the input first layer of a molecular graph will have information such as its energy, x/y/z location, and atom numbers. In the intermediate layers, they represent a specific pattern sensed by the network (such as high energy + hydrogen atom), even though the exact behaviors are yet to be human-interpretable. In such a manner, the neurons in the same position are expected to behave similarly, even though they correspond to different nodes.

These three observations shed light on how to close the performance gap between spiking GNNs are ANN-based GNNs. In the next section, we describe how the observations are used to build better spiking GNNs for graph classification.

## 4 Proposed Method

### 4.1 Overall Graph Classification Architecture

Many recent studies have tried to adapt SNN architectures into GNN tasks, however, they simply try to contact with only node classification tasks. In this work, we propose a spiking neural network specifically designed for graph classification tasks and show that it can be trained using spikes. We demonstrate the overall architecture of our graph classification model TAS-GNN in Figure 2. For each timestep, the input graphs are first translated into spike representations through the poisson encoder, then the message passing is done in spike format. After the combination phase in the GNN layer, the node features are once again binarized into spike format through passing the SNN layer. In the last layer, we perform an extra operation of aggregation and combination  on the spike features before passing the readout layer. The readout layer is essential to graph classification and is responsible for aggregating all the node embeddings in the graph into a single graph representation. A batch of graph embeddings is passed through a classification head that outputs logits for that timestep. To make the final prediction, we simply take the sum of logits from all timesteps and use softmax to obtain the class probabilities.

### 4.2 Topology-Aware Group-Adaptive Neurons

As discussed in Section 3, GNNs suffer from a huge gap in spike frequencies between neurons. As observed, there exist some patterns (Figure 5) that we can utilize to address the issue. One naive way of addressing the issue is to use learnable [49], or adaptive [4] threshold for each neuron. By adjusting the threshold, one can expect the neurons to naturally change, such that neurons under starvation will have lower thresholds to fire more often, and a few neurons with high firing rates will have higher thresholds to shift toward an ideal distribution.

Unfortunately, such an idea cannot be directly applied unless all the testset nodes are available at training time (i.e., transductive task). However, such a setting would be considered a data leak for graph classification, and would also lose the advantage SNNs have on lightweight inference.

Moreover, the number of nodes in a real-world dataset often ranges from at least thousands to several billions. Considering that GNNs often involve only a sub-million number of learnable parameters, storing such a large number of thresholds is considered too much overhead.

To address the aforementioned issues, we propose *topology-aware group adaptive neurons* (TAG), which partitions the neurons by their degrees. Note that $V_g$ denotes the node group to which the node is mapped, considering degree information. $S^{g_i}(t)$ and $V^{g_i}(t)$ represent the output spike and membrane potential of the $i$-th node in group $g$ at time $t$, respectively, as reformulated by Equation (4). We use $g$ to represent the unique degree distribution of the training sets. When an unseen node is encountered, we apply the initial threshold, as it has not been trained at all.

$$S^{g_i}(t) = \begin{cases} 1, & \text{if } V^{g_i}(t) \geq V_{th}^g(t-1) \\ 0, & \text{otherwise} \end{cases} \tag{8}$$

$$S^g(t) = \frac{1}{|V_g|} \sum_{i \in V_g} S^{g_i}(t) \tag{9}$$

$$V_{th}^g(t) = \gamma V_{th}^g(t-1) + (1-\gamma)S^g(t) \tag{10}$$

The major advantage of this scheme is that it is straightforward to put an unseen node or an unseen graph into a group at inference. To further consider intra-node deviation, we split the group into $F$ (number of features) neurons, which is a fixed parameter determined by the model architecture. For any unseen node, finding out its degree is trivial because visiting its neighbors is one of the fundamental requirements of graph data structures [26, 50, 36, 28]. Based on the observation ① from Section 3 that the neuron behavior is related to the degree, this will let neurons in the group collaboratively find an adequate threshold.

### 4.3 Reducing the Initial Threshold Sensitivity

The proposed Group-adaptive threshold scheme effectively reduces the spike frequency variation issue. However, we find that the adaptive neurons in the proposed TAG are sensitive to their initial thresholds. As depicted in Figure 3, the performance of the adaptive neurons can severely drop when the initial threshold value is not carefully tuned, which aligns with the findings from [4]. Moreover, manually tuning the initial thresholds individually is difficult because there are thousands of neuron groups.

To address the problem, we choose to learn the two parameters: the initial threshold per group ($V_{th}^g(0)$) and the decay rate ($\beta$). During training, we adopted the backpropagation algorithm [51, 10, 8] to update the value of $V_{th}^g(0)$

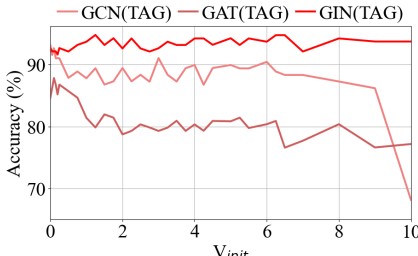

Figure 3: Sensitivity of neurons to its initial threshold.

with the gradients at time step t=1. This is done because $V_{th}^g(t)$ keeps updating with TAG Section 4.2 as time passes. During training, we also learn the decay rate ($\beta$) [16], which prevents the membrane voltage of neurons in low-degree nodes from leaking faster than it accumulates. For evaluation, we use the $V_{th}^g(0)$ values obtained during the training phase, adjusted for each group. The overall training procedure is in the Appendix.

## 5 Evaluation

### 5.1 Experiment Settings

We use a total of 5 graph datasets commonly used for benchmarking GNNs: MUTAG [9], PRO-TEINS [6], ENZYMES [6], NCI1 [47], and IMDB-Binary [54]. For the GNN layer in our architecture, we use 3 different designs, including GCN [27], GAT [45], and GIN [52]. The baselines include 3 works from SNN that are applicable to graph datasets: SpikingGNN [64], SpikeNet [32], and

Table 1: Performance comparison against baseline methods.

| Model | Method | MUTAG | PROTEINS | ENZYMES | NCI1 | IMDB-BINARY |
|---|---|---|---|---|---|---|
| | ANN [27] | 88.86 ± 5.48 | 77.81 ± 3.46 | 72.00 ± 4.37 | 76.42 ± 2.98 | 56.80 ± 4.80 |
| GCN | SpikingGNN [64] | 90.96 ± 3.99 | 74.39 ± 2.68 | 50.67 ± 4.91 | 73.41 ± 1.60 | 68.40 ± 2.96 |
| | SpikeNet [32] | 87.81 ± 5.60 | 74.75 ± 3.20 | 50.00 ± 3.33 | 73.92 ± 1.54 | 70.30 ± 2.17 |
| | PGNN [16] | 87.28 ± 5.87 | 77.36 ± 2.68 | 56.33 ± 3.17 | 76.52 ± 1.46 | 71.60 ± 2.17 |
| | TAS-GNN | **96.32** ± 3.10 (+5.35) | **77.45** ± 1.94 (+0.09) | **56.50** ± 3.87 (+0.17) | **77.81** ± 1.28 (+1.29) | **80.10** ± 2.49 (+8.50) |
| | ANN [45] | 83.04 ± 4.23 | 77.54 ± 3.22 | 59.67 ± 3.48 | 67.88 ± 3.00 | 54.50 ± 2.14 |
| GAT | SpikingGNN [64] | 78.71 ± 5.34 | 59.66 ± 0.21 | 29.17 ± 3.14 | 66.25 ± 1.77 | 50.00 ± 0.00 |
| | SpikeNet [32] | 78.22 ± 3.67 | 64.60 ± 3.24 | 51.67 ± 4.96 | 66.84 ± 1.60 | 50.00 ± 0.00 |
| | PGNN [16] | 82.49 ± 4.98 | 64.06 ± 2.37 | 39.50 ± 2.87 | 68.32 ± 1.49 | 50.00 ± 0.00 |
| | TAS-GNN | **96.32** ± 3.10 (+13.83) | **71.34** ± 3.03 (+6.74) | **52.33** ± 3.47 (+0.67) | **75.33** ± 2.41 (+7.01) | **77.90** ± 2.18 (+27.90) |
| | ANN [52] | 95.23 ± 5.61 | 78.79 ± 3.74 | 33.67 ± 4.66 | 79.17 ± 3.07 | 70.40 ± 4.14 |
| GIN | SpikingGNN [64] | 92.60 ± 4.41 | 77.81 ± 2.71 | 45.17 ± 5.01 | 70.29 ± 2.01 | 74.30 ± 1.47 |
| | SpikeNet [32] | 93.66 ± 4.62 | 78.43 ± 2.63 | 44.33 ± 3.98 | 74.77 ± 1.63 | **74.80** ± 2.74 |
| | PGNN [16] | 94.18 ± 4.84 | 79.16 ± 2.61 | 43.33 ± 5.45 | 75.38 ± 1.41 | 72.80 ± 4.63 |
| | TAS-GNN | **95.76** ± 3.47 (+1.58) | **80.32** ± 2.42 (+1.17) | **48.00** ± 4.01 (+2.83) | **77.52** ± 1.49 (+2.14) | 73.70 ± 3.11 (-1.10) |

†Did not converge

Table 2: Ablation study on the proposed method

| Model | Method | MUTAG | PROTEINS | ENZYMES | NCI1 | IMDB-BINARY |
|---|---|---|---|---|---|---|
| GCN | Baseline | 90.96 | 74.39 | 50.67 | 73.41 | 68.40 |
| | + TAG | 93.66 (+2.69) | 75.65 (+1.26) | 49.00 (-1.67) | 73.65 (+0.24) | 71.90 (+3.50) |
| | TAS-GNN (Proposed) | 96.32 (+5.35) | 77.45 (+3.06) | 56.50 (+5.83) | 77.81 (+4.40) | 80.10 (+11.70) |
| GAT | Baseline | 78.71 | 59.66 | 29.17 | 66.25 | 50.00 |
| | + TAG | 80.35 (+1.64) | 66.48 (+6.82) | 51.83 (+22.67) | 67.98 (+1.73) | 50.00 (+0.00) |
| | TAS-GNN (Proposed) | 96.32 (+17.60) | 71.34 (+11.68) | 52.33 (+23.16) | 75.33 (+9.08) | 77.90 (+27.90) |
| GIN | Baseline | 92.60 | 77.81 | 45.17 | 70.29 | 74.30 |
| | + TAG | 93.66 (+1.05) | 78.35 (+0.53) | 46.16 (+0.99) | 73.67 (+3.38) | 75.20 (+0.90) |
| | TAS-GNN (Proposed) | 95.76 (+3.16) | 80.32 (+2.51) | 48.00 (+2.83) | 77.52 (+7.23) | 73.70 (-0.60) |

PGNN [16]. Since this is the first SNN design to target graph classification, we apply minor modifications to each architecture, such as appending a readout layer. Note that SpikingGNN [64] was originally proposed for GCN, but we extend it to both GAT and GIN. More details on the experiment setting are included in the Appendix.

## 5.2 Results on Graph Classification

We compare TAS-GNN against prior works that adopt a spiking neural network to graph the dataset, shown in Table 1. We also report the performance of conventional ANN for comparison. In all but 2 cases, TAS-GNN outperforms the baselines by a noticeable margin. In the cases where TAS-GNN underperforms, the gaps are less than 1.1%p, smaller than the error bounds. In the opposite cases, the improvement is up to 27.90%p, showing a great amount of improvement.

An intriguing result is that TAS-GNN performs better than ANN-based GNNs in several cases. Improvements beyond the error bounds are found in MUTAG (GCN and GAT), NCI1 (GAT), and IMDB-BINARY (GCN and GAT). Note that the model architecture and the number of learnable parameters are the same in all methods. We believe this could come from the spiking neurons efficiently capturing the irregular connections over several timesteps, thereby showing an advantage over ANNs.

## 5.3 Ablation Study

In this section, we break down individual components of TAS-GNN and perform an ablation study, which is reported in Table 2. Starting from baseline implementation, which does not differentiate neurons used by each node, we apply TAG to show the effect of topology-aware group-adaptive neurons. Then, we add our learnable initial threshold scheme to complete TAS-GNN. The results show that TAG alone can improve the performance across all datasets and models. This means that uneven spike distribution caused by indegree variance is a general problem shared across different graph datasets, and simply grouping the nodes with similar indegree to share the same threshold helps alleviate this problem. Lastly, adding a learnable initial threshold scheme further boosts the accuracy in almost all cases, demonstrating its efficacy and stability.

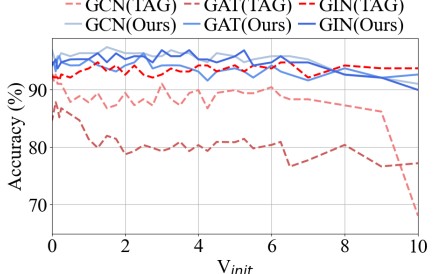

| Model | Method | $V_{init}$ | | | | | |
|---|---|---|---|---|---|---|---|
| | | 0.50 | 1.50 | 2.50 | 5.00 | 7.00 | 10.00 |
| GCN | TAG | 87.84 | 86.75 | 88.33 | 89.91 | 88.30 | 68.16 |
| | Ours | 95.79 | 97.37 | 96.32 | 95.79 | 95.23 | 90.99 |
| GAT | TAG | 85.70 | 81.96 | 80.35 | 80.85 | 77.72 | 77.19 |
| | Ours | 94.18 | 93.65 | 96.32 | 93.68 | 91.58 | 92.60 |
| GIN | TAG | 92.08 | 93.13 | 92.57 | 94.21 | 92.08 | 93.68 |
| | Ours | 94.18 | 94.74 | 95.76 | 93.68 | 94.71 | 89.94 |

Figure 4: Sensitivity study of neurons to its initial threshold.

## 5.4 Sensitivity Study

To validate our method's efficacy in alleviating the sensitivity of the initial threshold value, we perform a sensitivity study varying the values from 0.0 to 10.0. We compare our scheme against the TAG method, which also adaptively modulates the threshold during inference but does not learn it from training. Our method consistently performs indifferently to the initial threshold value, which means arduous search or tuning is unnecessary to achieve stable accuracy.

Table 3: Sensitivity study on threshold learning rate using MUTAG.

| Model | $\eta$ | | | | | |
|---|---|---|---|---|---|---|
| | 0.001 | 0.005 | 0.01 | 0.05 | 0.1 | 0.5 |
| GCN | 93.68 | 96.84 | 96.32 | 96.84 | 96.84 | 84.15 |
| GAT | 86.78 | 94.18 | 96.32 | 94.18 | 94.71 | 92.05 |
| GIN | 89.97 | 95.26 | 95.76 | 93.16 | 93.13 | 91.02 |

On the other hand, TAG is highly sensitive to the initial threshold and shows a performance gap up to 19.68%p except for GIN architecture, which is capturing structure well.

Since our scheme uses a learnable initial threshold, we also study its sensitivity for the learning rate, shown in Table 3. TAS-GNN performs best around $\eta = [0.005, 0.1]$, and starts to degrade for further increment or decrement. As denoted in the experimental setting, we use $\eta = 0.01$ as the default.

## 5.5 Additional Analysis

In this section, we give additional analysis on TAS-GNN by studying its spike frequency distribution. In Figure 5, we provide the same spike frequency visualization as done in Section 3, but using TAS-GNN. Unlike Figure 1, which showed severe starvation with most nodes not generating spikes, Figure 5a reveals that most nodes fire spikes, significantly alleviating the starvation problem. This is further illustrated Figure 5b, where most neurons have non-zero spike values and, what's more, meaningfully reflect the topology of the graph. For nodes with higher degrees, the spikes are more frequent (close to 5) due to having more incoming spikes from their neighbors. For GNNs, such information is essential to capture the global topology of the graph. This shows that our design of TAS-GNN faithfully reflects such information and can successfully propagate such information using spikes.

## 6 Related Works

**Graph Classification** Graph classification requires identifying the global characteristics of each graph and is commonly applied to domains such as bioinformatics [6], chemoinformatics [63], or social network analysis [21, 37]. Popular examples include the molecular classification of chemical compounds, proteins, or RNAs, where identifying the graph structural information is crucial. Due to the success of GNNs, [27, 45, 52, 57] Most GNNs use a message passing paradigm [20] that only aggregates local features. Thus, to obtain global features representing the entire graph, graph pooling [56] is often used. Global pooling summarizes the entire graph into a fixed-size graph embedding, which can be done by simply averaging or taking minimum or maximum values of the node-wise embeddings. Other variations replace such simple operations with neural networks [46, 33] or integrate sorting to selectively choose which node embeddings to include [60]. More advanced techniques such as hierarchical pooling utilze hierarchical information of graphs [40, 29, 18, 11] and usually show better representation learning. [60]

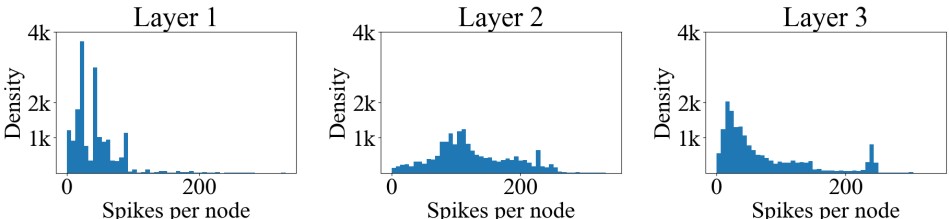

(a) Histogram plotting the distribution of total spikes counted over time for each node. X-axis denotes spike counts from each node, while y-axis denotes density of each bin.

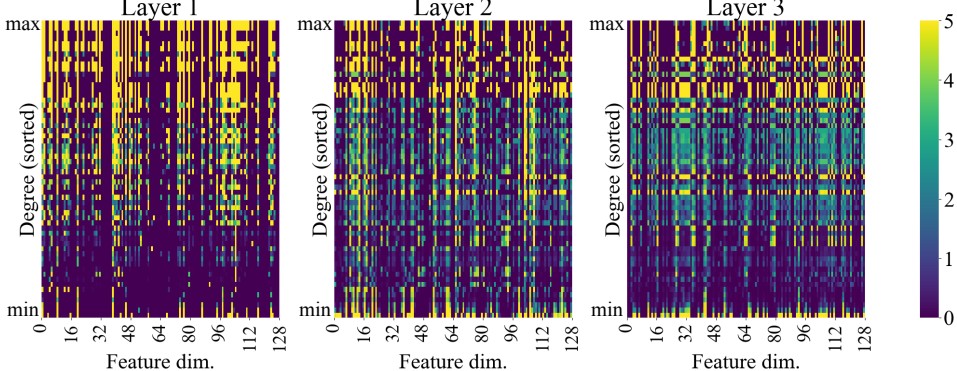

(b) Spike frequency visualization on TAS-GNN using each layer output. X-axis denotes feature dimension, while y-axis denotes nodes grouped and sorted by degree in descending order, top to bottom. Brighter spots denote higher frequency.

Figure 5: Analysis on spike frequency variation of GCN using IMDB-BINARY [54] dataset.

**Spiking Neural Networks** SNNs are a type of neural network where information is transmitted using spikes, similar to how biological neurons work. They use different neuron models for capturing spike signals effectively [23, 24] or adjusting parameters dynamically to compromise the accuracy [16, 49, 4, 34]. One major area of SNN research is converting traditional ANNs into SNNs by mapping ANN activation functions into spike signals [22, 41, 24, 42, 17]. Another focus is training SNNs directly using backpropagation, similar to ANNs, which involves using various techniques such as surrogate functions for backpropagation [43, 8] and adapting normalization techniques to SNNs [42, 12, 25, 62].

**SNN for Graphs** Previous attempts to apply SNNs to graph datasets have primarily focused on node-level classification tasks [59, 44, 64] and have not yet been extended to graph-level tasks. While [48] explored the application of spike training to Graph Attention Networks (GAT), it implemented the message passing phase after the spiking phase, which deviates from previous structures. Additionally, recent efforts have begun to integrate SNNs with other techniques for contrastive learning [31], particularly in dynamic graphs [55], to adopt collaboration between GNNs and SNNs.

## 7 Conclusion

In this paper, we explore the application of SNNs to graph neural networks for graph classification for the first time. After thoroughly analyzing the graph's uneven spike distribution, we identify that the degree of each node correlates to this phenomenon. To better accommodate such characteristics of graphs, we propose topology-aware group-adaptive neurons, which uses separate neurons for each degree group in the graph. In addition, we propose to learn the initial threshold and adaptively adjust the threshold simultaneously to reduce its sensitivity and facilitate training using spikes. Combined with the modified architecture for graph classification, we name our method TAS-GNN, and show that it outperforms existing works by a noticeable margin.

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

# A Appendix / supplemental material

## A.1 Limitation

Currently, our work is experimenting with the small-scale dataset for the graph classification that is generally used. However, we will extend our work into the large-scale dataset that could apply to the real. In addition, we will continue our future work for theoretical proof for updating the initial threshold that is fused with adaptative changes in the timestep.

## A.2 Code

The code which includes our implementation of this work is included in a zip archive of the supplementary material. The code is under Nvidia Source Code License-NC and GNU General Public License v3.0.

## A.3 Detailed Experiment Settings

**Dataset Details**   Given the diverse properties of graph datasets, we selected five datasets from the well-known TUDatasets, commonly used for graph classification. We compiled statistics for these datasets to briefly represent their key properties.

Table 4: Summary of datasets used in the study.

| Dataset | # Graphs | Avg. Nodes | # Nodes $(1^{st}graph)$ | Avg. Edges | # Edges $(1^{st}graph)$ | # Classes |
|---------|----------|------------|-------------------------|------------|-------------------------|-----------|
| MUTAG [9] | 188 | 17.93 | 17 | 19.79 | 38 | 2 |
| PROTEINS [6] | 1113 | 39.06 | 42 | 72.82 | 162 | 2 |
| ENZYMES [6] | 600 | 32.6 | 37 | 62.1 | 168 | 6 |
| NCI1 [47] | 4110 | 29.87 | 21 | 32.30 | 42 | 2 |
| IMDB-BINARY [54] | 1000 | 19.77 | 20 | 96.53 | 146 | 2 |

**Network Architecture**   In this work, we consider the following three GNN architectures where the distinctions lie in their update rules:

- Graph Convolution Network [27] (GCN): $h_i^{(l+1)} = \sigma(\sum_{j \in \mathcal{N}(i) \bigcup \{i\}} \frac{Wh_j^{(l)}}{\sqrt{|N(i)||N(j)|}})$, where $\phi(\cdot)$ is replaced by affine transformation $W$ followed by nonlinearity $\sigma$.

- Graph Attention Network [45] (GAT): $h_i^{(l+1)} = \alpha_{i,i}Wh_i^{(l)} + \sum_{j \in \mathcal{N}(i)} \alpha_{ij}Wh_j^{(l)}$, where $\alpha_{ij}$ is the normalized attention score between node $i$ and $j$.

- Graph Isomorphism Network [52] (GIN): $h_i^{(l+1)} = MLP((1 + \epsilon)h_i^{(l)} + \sum_{j \in \mathcal{N}(i)} h_j^{(l)})$, where $\epsilon$ is a learnable constant.

For the GCN layers, 128 dimensions were used for hidden dimensions, and GAT layers were used for 4 multi-head attentions. GIN was used for 2-MLP layers for the above equation.

**Experiment Settings**   We trained and evaluated our models using 10-fold cross-validation for all datasets. Note that the IMDB-BINARY dataset lacks inherent features, so we constructed features using the node degrees for the GNN layer. Additionally, we did not apply any multiplier to adjust the width of the sigmoid function. The details of our evaluation procedure are outlined below. Our experiment was evaluated on a single RTX-4090 GPU for the full batch GNN training.

- Epochs: 1000
- Surrogate function: $\sigma(x) = \frac{1}{1+e^{-x}}$
- Learning rate($\eta$): 0.01 (for main table)
- Optimizer: Adamw
- Loss function: Cross entropy
- Adaptive step size($\gamma$): 0.2

## A.4 Analysis on Spike Frequency

We provide additional figures that we referenced on Section 3. Appendix A.4 shows MUTAG, PROTEINS, ENZYMES, NCI1 dataset total spike histogram bins.

## A.5 Overall training procedure

As referred on Section 4 our TAG method and overall updating initial values of group threshold is reffed on Algorithm 1. Note that our initial group values updated after timestep T.

---

**Algorithm 1** Updataing $V_{th}^g(0)$ procedure

---

1: **Inputs:** Initial start points of threshold $V_{init}$, graph's vertex feature $X \in R^{VXF}$, learning rate for training $\eta$, total time step $T$, $l$-th layer's threshold $V_{th}^{(l)}$, $l$-th layer's GNN layer $GNN^{(l)}$, true label $Y$,
2: **Initialize:** $V_{th}^g(0) = [V_{init}, ... V_{init}]$      ▷ Initialize all of the g threshold groups with initial values
3: **for** $ep = 1$ to $epochs$ **do**
4:    **for** $t = 1$ to $T$ **do**
5:      $X = PoissonEncoder(X)$      ▷ Binarize first input layer with Poisson encoder
6:      **for** $l = 1$ to $L$ **do**
7:        **for** $g$ in group $G$ **do**
8:          $X^{g,(l)} = GNN^{(l)}(X^{g,(l)})$      ▷ Operate by GCN, GAT, GIN architectures
9:          **for** $i = 1$ to $|V_g|$ **do**
10:            $X^{g_i,(l)} = S^{g_i,(l)}(t) = SNN^{(l)}(X^{g_i,(l)})$    ▷ $X^{g_i,(l)}$ represents $i$-th row of $X^{g,(l)}$
11:            $S^{g,(l)}(t) = \frac{1}{|V_g|} \sum_{i \in V_g} S^{g_i}(t)$
12:          **end for**
13:          $V_{th}^{g,(l)}(t) = \gamma V_{th}^{g,(l)}(t-1) + (1-\gamma)S^{g,(l)}(t)$ ▷ Update threshold through TAG Equation (10)
14:        **end for**
15:      **end for**
16:      $O^t \leftarrow FC(POOL(GNN(X^{(L)}))) + O^{t-1}$
17:    **end for**
18:    $V_{th}(0) = V_{th}(0) - \eta \nabla_{V_{th}(0)} \mathcal{L}(O^{t=1}, Y)$
19: **end for**

---

## A.6 Sensitivity Study on Degree Group

Our experiments were conducted on a number of degree groups. Please refer Table 5 for the sensitivity depending on the number of degree groups. Please note that the optimal values of the degree groups are different depending on the graph datasets. We reported to the max degree group setting that unseen nodes will use the initial values $V_{init}$ that represents the $V_{th}(0)$ that does not trained at all.

## A.7 Sensitivity Study on Learning Rate

Our experiments were conducted under various learning rate conditions $\eta \in [0.001, 0.5]$ to assess their impact. As reported in Table 3 for the MUTAG dataset, we also present results for the PROTEINS, ENZYMES, NCI1, and IMDB-BINARY datasets across GCN, GAT, and GIN architectures. Our model's ability to learn $V_{init}$ demonstrates a sensitivity to learning rate similar to other ANN models. We found that the optimal performance was achieved at a learning rate of $\eta = 0.01$.

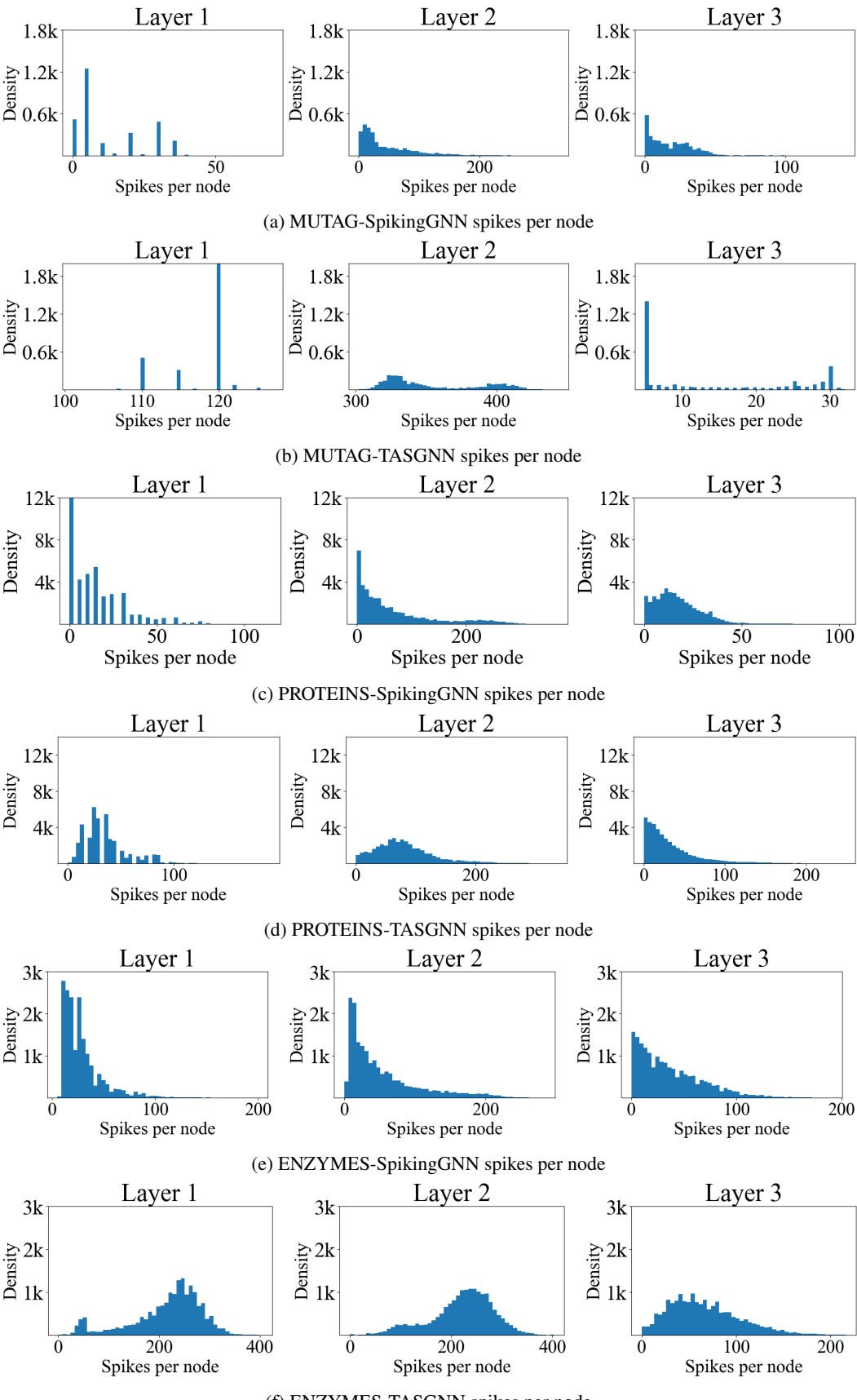

(a) MUTAG-SpikingGNN spikes per node

(b) MUTAG-TASGNN spikes per node

(c) PROTEINS-SpikingGNN spikes per node

(d) PROTEINS-TASGNN spikes per node

(e) ENZYMES-SpikingGNN spikes per node

(f) ENZYMES-TASGNN spikes per node

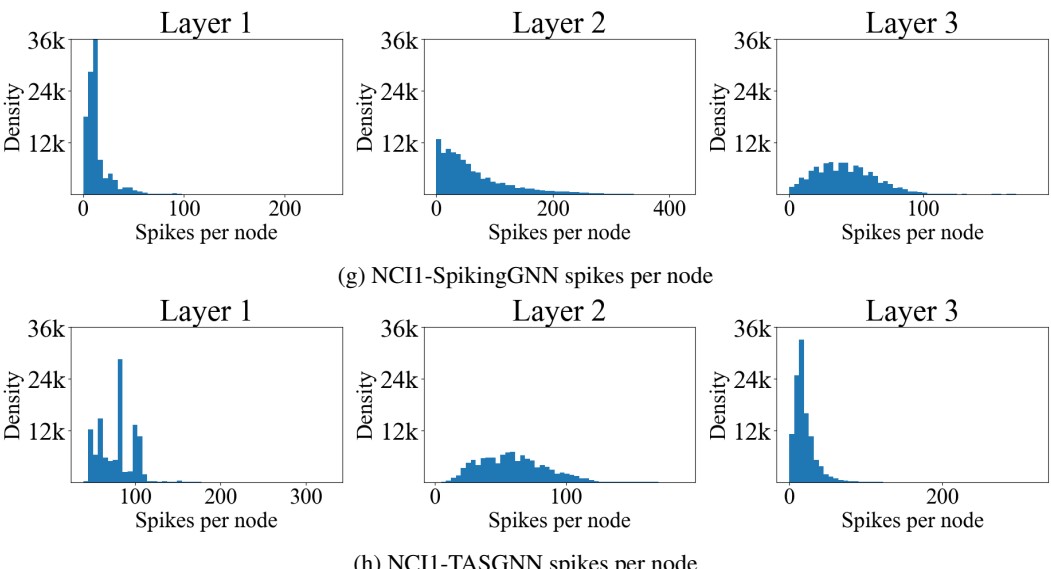

(g) NCI1-SpikingGNN spikes per node

(h) NCI1-TASGNN spikes per node

Figure 6: Histogram plotting distribution of total spikes counted over time for each node. X-axis denotes spike counts from each node, while y-axis denotes density of each bin.


Table 5: Comparison on using different number of degree group

| Dataset | #Degree Group | GCN | GAT | GIN |
|---|---|---|---|---|
| MUTAG | 1 | 87.81 | 80.88 | 94.71 |
| | 2 | 96.84 | 87.78 | 96.32 |
| | 3 | 93.10 | 95.79 | 95.79 |
| | 4(max) | 96.32 | 96.32 | 95.76 |
| PROTEINS | 1 | 78.89 | 64.33 | 78.89 |
| | 2 | 78.98 | 63.88 | 78.98 |
| | 5 | 75.83 | 67.39 | 75.83 |
| | 10 | 77.45 | 69.55 | 77.45 |
| | 15 | 77.99 | 70.54 | 77.99 |
| | 17(max) | 77.45 | 71.34 | 80.32 |
| ENZYMES | 1 | 58.33 | 41.33 | 45.17 |
| | 2 | 56.50 | 40.50 | 44.33 |
| | 5 | 52.00 | 45.00 | 41.50 |
| | 10(max) | 56.50 | 52.33 | 48.00 |
| NCI1 | 1 | 75.74 | 67.86 | 73.82 |
| | 2 | 75.77 | 68.08 | 75.06 |
| | 3 | 77.86 | 72.48 | 76.86 |
| | 4 | 77.81 | 74.26 | 76.74 |
| | 5(max) | 77.81 | 75.33 | 77.52 |
| IMDB-BINARY | 1 | 71.70 | 50.00 | 74.60 |
| | 2 | 70.40 | 50.30 | 72.90 |
| | 5 | 69.30 | 56.80 | 71.00 |
| | 10 | 66.70 | 56.40 | 66.70 |
| | 20 | 64.00 | 61.30 | 66.20 |
| | 50 | 65.99 | 64.51 | 65.55 |
| | 65(max) | 80.10 | 77.90 | 73.70 |


Table 6: Extended sensitivity study on threshold learning rate.

| Dataset | Model | $\eta$ 0.001 | 0.005 | 0.01 | 0.05 | 0.1 | 0.5 |
|---------|-------|-------|-------|------|------|-----|-----|
| MUTAG | GCN | 93.68 | 96.84 | 96.32 | 96.84 | 96.84 | 84.15 |
| | GAT | 86.78 | 94.18 | 96.32 | 94.18 | 94.71 | 92.05 |
| | GIN | 89.97 | 95.26 | 95.76 | 93.16 | 93.13 | 91.02 |
| PROTEINS | GCN | 75.11 | 76.82 | 77.45 | 77.36 | 76.82 | 65.67 |
| | GAT | 64.14 | 70.35 | 71.34 | 73.23 | 74.93 | 70.53 |
| | GIN | 77.72 | 79.07 | 80.32 | 78.17 | 76.55 | 75.65 |
| ENZYMES | GCN | 45.00 | 51.17 | 56.50 | 56.83 | 54.67 | 29.17 |
| | GAT | 32.00 | 45.00 | 52.33 | 55.83 | 42.67 | 34.33 |
| | GIN | 37.33 | 44.33 | 48.00 | 35.17 | 31.33 | 29.33 |
| NCI1 | GCN | 73.87 | 77.37 | 77.81 | 80.07 | 78.81 | 66.95 |
| | GAT | 66.93 | 73.31 | 75.33 | 76.06 | 73.48 | 66.69 |
| | GIN | 72.80 | 76.57 | 77.52 | 70.54 | 69.05 | 64.94 |
| IMDB-Binary | GCN | 78.90 | 79.90 | 80.10 | 80.50 | 80.60 | 73.60 |
| | GAT | 74.80 | 75.80 | 77.90 | 75.60 | 75.90 | 75.30 |
| | GIN | 74.10 | 73.00 | 73.70 | 75.40 | 74.70 | 73.60 |

- The authors are encouraged to create a separate "Limitations" section in their paper.
- The paper should point out any strong assumptions and how robust the results are to violations of these assumptions (e.g., independence assumptions, noiseless settings, model well-specification, asymptotic approximations only holding locally). The authors should reflect on how these assumptions might be violated in practice and what the implications would be.
- The authors should reflect on the scope of the claims made, e.g., if the approach was only tested on a few datasets or with a few runs. In general, empirical results often depend on implicit assumptions, which should be articulated.
- The authors should reflect on the factors that influence the performance of the approach. For example, a facial recognition algorithm may perform poorly when image resolution is low or images are taken in low lighting. Or a speech-to-text system might not be used reliably to provide closed captions for online lectures because it fails to handle technical jargon.
- The authors should discuss the computational efficiency of the proposed algorithms and how they scale with dataset size.
- If applicable, the authors should discuss possible limitations of their approach to address problems of privacy and fairness.
- While the authors might fear that complete honesty about limitations might be used by reviewers as grounds for rejection, a worse outcome might be that reviewers discover limitations that aren't acknowledged in the paper. The authors should use their best judgment and recognize that individual actions in favor of transparency play an important role in developing norms that preserve the integrity of the community. Reviewers will be specifically instructed to not penalize honesty concerning limitations.

3. **Theory Assumptions and Proofs**

Question: For each theoretical result, does the paper provide the full set of assumptions and a complete (and correct) proof?

Answer: [NA]

Justification: Our work does not include theoretical results.

Guidelines:

- The answer NA means that the paper does not include theoretical results.
- All the theorems, formulas, and proofs in the paper should be numbered and cross-referenced.
- All assumptions should be clearly stated or referenced in the statement of any theorems.

- The proofs can either appear in the main paper or the supplemental material, but if they appear in the supplemental material, the authors are encouraged to provide a short proof sketch to provide intuition.
- Inversely, any informal proof provided in the core of the paper should be complemented by formal proofs provided in appendix or supplemental material.
- Theorems and Lemmas that the proof relies upon should be properly referenced.

4. **Experimental Result Reproducibility**

Question: Does the paper fully disclose all the information needed to reproduce the main experimental results of the paper to the extent that it affects the main claims and/or conclusions of the paper (regardless of whether the code and data are provided or not)?

Answer: [Yes]

Justification: We provided our codes that able to reproduce our model's result.

Guidelines:

- The answer NA means that the paper does not include experiments.
- If the paper includes experiments, a No answer to this question will not be perceived well by the reviewers: Making the paper reproducible is important, regardless of whether the code and data are provided or not.
- If the contribution is a dataset and/or model, the authors should describe the steps taken to make their results reproducible or verifiable.
- Depending on the contribution, reproducibility can be accomplished in various ways. For example, if the contribution is a novel architecture, describing the architecture fully might suffice, or if the contribution is a specific model and empirical evaluation, it may be necessary to either make it possible for others to replicate the model with the same dataset, or provide access to the model. In general. releasing code and data is often one good way to accomplish this, but reproducibility can also be provided via detailed instructions for how to replicate the results, access to a hosted model (e.g., in the case of a large language model), releasing of a model checkpoint, or other means that are appropriate to the research performed.
- While NeurIPS does not require releasing code, the conference does require all submissions to provide some reasonable avenue for reproducibility, which may depend on the nature of the contribution. For example
  (a) If the contribution is primarily a new algorithm, the paper should make it clear how to reproduce that algorithm.
  (b) If the contribution is primarily a new model architecture, the paper should describe the architecture clearly and fully.
  (c) If the contribution is a new model (e.g., a large language model), then there should either be a way to access this model for reproducing the results or a way to reproduce the model (e.g., with an open-source dataset or instructions for how to construct the dataset).
  (d) We recognize that reproducibility may be tricky in some cases, in which case authors are welcome to describe the particular way they provide for reproducibility. In the case of closed-source models, it may be that access to the model is limited in some way (e.g., to registered users), but it should be possible for other researchers to have some path to reproducing or verifying the results.

5. **Open access to data and code**

Question: Does the paper provide open access to the data and code, with sufficient instructions to faithfully reproduce the main experimental results, as described in supplemental material?

Answer: [Yes]

Justification: We provide our codes that are able to reproduce our full experiments.

Guidelines:

- The answer NA means that paper does not include experiments requiring code.
- Please see the NeurIPS code and data submission guidelines (`https://nips.cc/public/guides/CodeSubmissionPolicy`) for more details.

- While we encourage the release of code and data, we understand that this might not be possible, so "No" is an acceptable answer. Papers cannot be rejected simply for not including code, unless this is central to the contribution (e.g., for a new open-source benchmark).
- The instructions should contain the exact command and environment needed to run to reproduce the results. See the NeurIPS code and data submission guidelines (`https://nips.cc/public/guides/CodeSubmissionPolicy`) for more details.
- The authors should provide instructions on data access and preparation, including how to access the raw data, preprocessed data, intermediate data, and generated data, etc.
- The authors should provide scripts to reproduce all experimental results for the new proposed method and baselines. If only a subset of experiments are reproducible, they should state which ones are omitted from the script and why.
- At submission time, to preserve anonymity, the authors should release anonymized versions (if applicable).
- Providing as much information as possible in supplemental material (appended to the paper) is recommended, but including URLs to data and code is permitted.

6. **Experimental Setting/Details**

Question: Does the paper specify all the training and test details (e.g., data splits, hyper-parameters, how they were chosen, type of optimizer, etc.) necessary to understand the results?

Answer: [Yes]

Justification: We wrote experiment setting in the experiment settings including GNN layers, hyperparameter for the hidden dimension, and learning rate of the whole dataset. Also, we wrote epochs and dataset we split was used by 10 fold CV for our evaluations.

Guidelines:

- The answer NA means that the paper does not include experiments.
- The experimental setting should be presented in the core of the paper to a level of detail that is necessary to appreciate the results and make sense of them.
- The full details can be provided either with the code, in appendix, or as supplemental material.

7. **Experiment Statistical Significance**

Question: Does the paper report error bars suitably and correctly defined or other appropriate information about the statistical significance of the experiments?

Answer: [Yes]

Justification: We reported error of confidence level in the main table.

Guidelines:

- The answer NA means that the paper does not include experiments.
- The authors should answer "Yes" if the results are accompanied by error bars, confidence intervals, or statistical significance tests, at least for the experiments that support the main claims of the paper.
- The factors of variability that the error bars are capturing should be clearly stated (for example, train/test split, initialization, random drawing of some parameter, or overall run with given experimental conditions).
- The method for calculating the error bars should be explained (closed form formula, call to a library function, bootstrap, etc.)
- The assumptions made should be given (e.g., Normally distributed errors).
- It should be clear whether the error bar is the standard deviation or the standard error of the mean.
- It is OK to report 1-sigma error bars, but one should state it. The authors should preferably report a 2-sigma error bar than state that they have a 96% CI, if the hypothesis of Normality of errors is not verified.

- For asymmetric distributions, the authors should be careful not to show in tables or figures symmetric error bars that would yield results that are out of range (e.g. negative error rates).
- If error bars are reported in tables or plots, The authors should explain in the text how they were calculated and reference the corresponding figures or tables in the text.

8. **Experiments Compute Resources**

Question: For each experiment, does the paper provide sufficient information on the computer resources (type of compute workers, memory, time of execution) needed to reproduce the experiments?

Answer: [Yes]

Justification: It refers to the appendix for experimental settings.

Guidelines:

- The answer NA means that the paper does not include experiments.
- The paper should indicate the type of compute workers CPU or GPU, internal cluster, or cloud provider, including relevant memory and storage.
- The paper should provide the amount of compute required for each of the individual experimental runs as well as estimate the total compute.
- The paper should disclose whether the full research project required more compute than the experiments reported in the paper (e.g., preliminary or failed experiments that didn't make it into the paper).

9. **Code Of Ethics**

Question: Does the research conducted in the paper conform, in every respect, with the NeurIPS Code of Ethics https://neurips.cc/public/EthicsGuidelines?

Answer: [Yes]

Justification: Research conducted in the paper conforms, in every respect, with the NeurIPS Code of Ethics

Guidelines:

- The answer NA means that the authors have not reviewed the NeurIPS Code of Ethics.
- If the authors answer No, they should explain the special circumstances that require a deviation from the Code of Ethics.
- The authors should make sure to preserve anonymity (e.g., if there is a special consideration due to laws or regulations in their jurisdiction).

10. **Broader Impacts**

Question: Does the paper discuss both potential positive societal impacts and negative societal impacts of the work performed?

Answer: [Yes]

Justification: SNN would be one of the breakthrough idea in respect of energy consumption.

Guidelines:

- The answer NA means that there is no societal impact of the work performed.
- If the authors answer NA or No, they should explain why their work has no societal impact or why the paper does not address societal impact.
- Examples of negative societal impacts include potential malicious or unintended uses (e.g., disinformation, generating fake profiles, surveillance), fairness considerations (e.g., deployment of technologies that could make decisions that unfairly impact specific groups), privacy considerations, and security considerations.
- The conference expects that many papers will be foundational research and not tied to particular applications, let alone deployments. However, if there is a direct path to any negative applications, the authors should point it out. For example, it is legitimate to point out that an improvement in the quality of generative models could be used to generate deepfakes for disinformation. On the other hand, it is not needed to point out that a generic algorithm for optimizing neural networks could enable people to train models that generate Deepfakes faster.

- The authors should consider possible harms that could arise when the technology is being used as intended and functioning correctly, harms that could arise when the technology is being used as intended but gives incorrect results, and harms following from (intentional or unintentional) misuse of the technology.
- If there are negative societal impacts, the authors could also discuss possible mitigation strategies (e.g., gated release of models, providing defenses in addition to attacks, mechanisms for monitoring misuse, mechanisms to monitor how a system learns from feedback over time, improving the efficiency and accessibility of ML).

11. **Safeguards**

Question: Does the paper describe safeguards that have been put in place for responsible release of data or models that have a high risk for misuse (e.g., pretrained language models, image generators, or scraped datasets)?

Answer: [NA]

Justification: Our paper poses no such risks for high risk for misuse.

Guidelines:

- The answer NA means that the paper poses no such risks.
- Released models that have a high risk for misuse or dual-use should be released with necessary safeguards to allow for controlled use of the model, for example by requiring that users adhere to usage guidelines or restrictions to access the model or implementing safety filters.
- Datasets that have been scraped from the Internet could pose safety risks. The authors should describe how they avoided releasing unsafe images.
- We recognize that providing effective safeguards is challenging, and many papers do not require this, but we encourage authors to take this into account and make a best faith effort.

12. **Licenses for existing assets**

Question: Are the creators or original owners of assets (e.g., code, data, models), used in the paper, properly credited and are the license and terms of use explicitly mentioned and properly respected?

Answer: [Yes]

Justification: We reported owners of assets used in the paper in the Appendix

Guidelines:

- The answer NA means that the paper does not use existing assets.
- The authors should cite the original paper that produced the code package or dataset.
- The authors should state which version of the asset is used and, if possible, include a URL.
- The name of the license (e.g., CC-BY 4.0) should be included for each asset.
- For scraped data from a particular source (e.g., website), the copyright and terms of service of that source should be provided.
- If assets are released, the license, copyright information, and terms of use in the package should be provided. For popular datasets, `paperswithcode.com/datasets` has curated licenses for some datasets. Their licensing guide can help determine the license of a dataset.
- For existing datasets that are re-packaged, both the original license and the license of the derived asset (if it has changed) should be provided.
- If this information is not available online, the authors are encouraged to reach out to the asset's creators.

13. **New Assets**

Question: Are new assets introduced in the paper well documented and is the documentation provided alongside the assets?

Answer: [Yes]

Justification: Considering our implemtation code is our asset, our work provides necessary license and documents for further usage.

Guidelines:

- The answer NA means that the paper does not release new assets.
- Researchers should communicate the details of the dataset/code/model as part of their submissions via structured templates. This includes details about training, license, limitations, etc.
- The paper should discuss whether and how consent was obtained from people whose asset is used.
- At submission time, remember to anonymize your assets (if applicable). You can either create an anonymized URL or include an anonymized zip file.

14. **Crowdsourcing and Research with Human Subjects**

Question: For crowdsourcing experiments and research with human subjects, does the paper include the full text of instructions given to participants and screenshots, if applicable, as well as details about compensation (if any)?

Answer: [NA]

Justification: Our work does not involve crowdsourcing nor research with human subjects.

Guidelines:

- The answer NA means that the paper does not involve crowdsourcing nor research with human subjects.
- Including this information in the supplemental material is fine, but if the main contribution of the paper involves human subjects, then as much detail as possible should be included in the main paper.
- According to the NeurIPS Code of Ethics, workers involved in data collection, curation, or other labor should be paid at least the minimum wage in the country of the data collector.

15. **Institutional Review Board (IRB) Approvals or Equivalent for Research with Human Subjects**

Question: Does the paper describe potential risks incurred by study participants, whether such risks were disclosed to the subjects, and whether Institutional Review Board (IRB) approvals (or an equivalent approval/review based on the requirements of your country or institution) were obtained?

Answer: [NA]

Justification: Our work does not require IRB approvals and does not involve human subjects.

Guidelines:

- The answer NA means that the paper does not involve crowdsourcing nor research with human subjects.
- Depending on the country in which research is conducted, IRB approval (or equivalent) may be required for any human subjects research. If you obtained IRB approval, you should clearly state this in the paper.
- We recognize that the procedures for this may vary significantly between institutions and locations, and we expect authors to adhere to the NeurIPS Code of Ethics and the guidelines for their institution.
- For initial submissions, do not include any information that would break anonymity (if applicable), such as the institution conducting the review.

