# OpenReview forum: "TAS-GNN: Topology-Aware Spiking Graph Neural Networks for Graph Classification"
_NeurIPS.cc/2024/Conference — Submitted to NeurIPS 2024_

### Official Review · Reviewer_5K64 · 2024-07-02

**Soundness:** 3
**Presentation:** 2
**Contribution:** 2
**Rating:** 6
**Confidence:** 4

**Summary:**

There's a large performance gap for graph tasks, especially graph classification tasks, between the spiking neural networks and artificial neural networks. The authors proposes the problems as the neuron's under starvation and illustrated the reason of the problem. To solve the problem, TAS-GNN was proposed.

The main contributions of the paper are as follows:
1: Starvation problem of spiking neurone in GNNs in graph classification tasks are identified.

2: A strategy was proposed to address the spike frequency deviations on the basis of the correlation between graph topology and spike frequency patterns.

The authors conduct experiments on 5 popular datasets and use several different designs of GNN layer. The results show competitive potential of the TAS-GNN.

**Strengths:**

1:This is a well-written paper, from the formulation of the problem to the solution. The author's motivation for the use of graph topology is clear.

2:The method of using topology-awaregroup-adaptive neurons shows competitive results compared with other baselines. The ablation study makes the result more persuasive.

3: The Figures in the paper are quite straightforward, easy to follow.

**Weaknesses:**

1: The name of the paper is "Topology-Aware Spiking Graph Neural Networks". However, as I can tell the only graph topology used in the method is nodes degree, which is used to group the neurons. I wonder if it is appropriate to name it as "topology aware", or the author can explain it more.

2: The analysis regarding the performance of the method is lack of discussion. For instance, in some datasets, such as MUTAG and IMDB-Binary, the proposed method achieve quite competitive results while in PROTEINS it doesn't. It's betted to explain what cause the phenomenon, like the characteristics of the datasets? Also, in table 2, the results of GAT and GAT+TAG in IMDB-Binary are the same. It's better to make an explanation about them.

3: There're several typos and basic grammar mistakes in the paper that will affect the presentation of the paper. In line 120 " and apply is to"; The sentence in line 123 is hard to understand

**Questions:**

1: In section 3 the authors mentioned the hypothesis that the phenomenon mentioned above is caused by the topology of the real-world graphs. What motivates you to have the hypothesis?

---

> ### Author Rebuttal · Authors · 2024-08-07
>
> Thank you for acknowledging our contributions, along with positive and constructive feedbacks. We respond to the comments as below.
>
>
> ### **W1. Gap between graph topology and node degree**
>
> We used node degree information for one of the representative graph topology properties. As the reviewer mentioned, degree information is not the entirety of topology information. Thus, we will refine our claim to reflect that we used degree information instead of topology. The reason why we initially used the term topology is that we regarded degree information as a representative feature of graph topology and thought topology would be better understood than degree to readers. For instance, several papers [1, 2, 3, 4] have utilized degree as a core information representing the topology.
>
> [1] Bounova, Gergana, et al. "Overview of Metrics and Their Correlation Patterns for Multiple-Metric Topology Analysis on Heterogeneous Graph Ensembles." Physical Review E, 2012.
> [2] Tangmunarunkit, Hongsuda, et al. "Network Topology Generators: Degree-Based vs. Structural." SIGCOMM Computer Communication Review, 2002.
> [3] Zhang, X., et al. "On Degree Based Topological Properties of Two Carbon Nanotubes." Polycyclic Aromatic Compounds, 2020.
> [4] Zhou, Shi, and Raúl J. Mondragón. "Accurately Modeling the Internet Topology." Physical Review E, 2004.
>
>
> ### **W2. Lack of analysis for performance**
>
> * **Small performance gain on PROTEINS dataset**
>
>
> Before analyzing the performance gains, please allow us to clarify that we believe our performance in PROTEINS is also competitive as it achieves significant improvements (up to 6.74%) over the SNN baselines. Typically, it is widely perceived as a normal behavior for current SNNs to achieve slightly lower performance compared to ANNs (e.g., SNN transformers [5, 6, 7]) despite their energy efficiency. Thus, the goal is usually to narrow the gap between the ANNs and SNNs, and comparisons are often done among other SNNs. It might seem contradictory because many of the TAS-GNN results outperform ANNs in Table 1. On this, we are genuinely excited about finding out a case where SNNs beat their ANN counterparts, although the exact reason is yet to be investigated.
>
> Having said that, the performance improvement in PROTEINS is relatively smaller than on other datasets since it did not beat ANNs. We would like to provide our analysis on this. We believe the reason is related to the spike diversity in the last-layer neurons. We could observe that TAS-GNN was not sufficiently diverse in the last-layer spike histogram (Figure 6 in the Appendix) on the PROTEINS dataset. In contrast, in the other datasets (MUTAG, ENZYMES, IMDB-BINARY), last-layer spikes were effectively more diverse than those of SpikingGNN. From this, it might be possible to achieve higher performance on the PROTEINS dataset if we could further increase the spike diversity in the last layer.
>
> [5] Zhou, Zhaokun, et al. “Spikformer: When Spiking Neural Network Meets Transformer.” ICLR, 2023.
> [6] Zhou, Zhaokun, et al. “Spikformer v2: Join the High Accuracy Club on ImageNet with an SNN Ticket.” arXiv:2401.02020, 2024.
> [7] Yao, Man, et al. "Spike-driven transformer." NeurIPS, 2024.
>
> * **The same performance for GAT, GAT+TAG method accuracy**
>
> GAT and GAT+TAG both diverge on IMDB-Binary (50% accuracy on binary classification). This issue may relate to the dataset's and GAT architecture’s properties. For instance, the IMDB-BINARY dataset has no node features, meaning the only available information is the degree information used when passing through the GNN layers. Specifically, the absence of node features becomes significant in the Graph Attention Network (GAT) layer, which focuses on smoothing degree information through learnable edge weights (i.e., attention). This is why using TAG on the IMDB-BINARY dataset could be ineffective; the attention mechanism diminishes the importance of the degree information.
>
> ### **W3. Writing mistakes on paper**
>
>  Thank you for suggesting the grammar mistakes in our paper. We appreciate the detailed review and will ensure that all errors, including the ones mentioned by the reviewer on lines 120 and 123, are corrected. We will carefully review the entire document to address any other potential issues.
>
> * line 120: apply is to → apply them to
> * line 123: SNN layer, consist → SNN layer. It consists
> * line 213: keeps updating with → updating with
> * line 288: hiearchical , utilze → hierarchical, utilize
> * line 300: explored → exploring
>
>
> ### **Q1. How did we get the motivation by topology information by observing the phenomenon in section 3**
>
> We were motivated by the pattern similarity between real-world graph distributions and the spike density distributions. Many real-world graphs are popular for their power-law distribution [8, 9], which indicates that there exists a few high-degree nodes with the majority of low-degree nodes. When we observed the starvation problem in Figure 1 (a), we realized that the pattern resembles that of the well-known degree distributions. This had led us to the hypothesis, which was later validated in Fig1(b).
>
>
>
> [8] Jure Leskovec et al. “Graphs over time: densification laws, shrinking diameters and possible explanations.” KDD. 2005.
> [9] Jure Leskovec et al. 2007. “Graph evolution: Densification and shrinking diameters.” ACM Trans. Knowl. Discov. Data, 2007.

---

> > ### Comment · Reviewer_5K64 · 2024-08-09
> > **Response to Author Rebuttal**
> >
> > I would like to thank the authors for their detailed discussion and for addressing my concerns. As mentioned by one of the authors, the term 'graph topology' instead of 'degree' can be misleading. I hope the authors will further consider refining this expression. I've increased my score. Good luck!

---

> > > ### Author Response · Authors · 2024-08-11
> > >
> > > Thank you! We sincerely appreciate the reviewer for the feedback. We will make sure to address the issue in the paper. If there are any additional points the reviewer would like to clarify, please let us know. We will be more than happy to address them.

---

### Official Review · Reviewer_icJz · 2024-07-05

**Soundness:** 3
**Presentation:** 2
**Contribution:** 3
**Rating:** 7
**Confidence:** 5

**Summary:**

This paper primarily discusses integrating Spiking Neural Networks (SNNs) into Graph Neural Networks (GNNs) to address several key challenges in graph classification tasks. Specifically, the paper proposes a new method called TAS-GNN (Topology-Aware Spiking Graph Neural Networks) which leverages the topology of graphs to improve the performance of spiking neural networks in graph classification tasks.

**Strengths:**

（1）The authors clearly articulate the performance gap between existing Graph Neural Networks (GNNs) and Spiking Neural Networks (SNNs) in graph classification tasks.
（2）The authors conduct an in-depth analysis of the performance degradation of spiking neural networks in graph classification tasks and introduce the "neuron starvation" problem.
（3）The authors propose topology-aware group-adaptive neurons (TAG) based on the graph's topology, a novel approach that helps address the neuron starvation issue.
（4）The authors provide a detailed description of how to convert input graphs into spike representations, perform message passing, and classify the graphs.
（5）The authors validate the method's generalizability and effectiveness by using multiple public datasets (such as MUTAG, PROTEINS, ENZYMES, NCI1, IMDB-BINARY) in the experimental section.

**Weaknesses:**

（1）The authors mention several application areas and challenges, but the references and comparisons to existing literature are not sufficiently comprehensive.
（2）Although the methodology section describes the main steps, it lacks detailed descriptions of some key aspects such as threshold initialization and the specific training process.
（3）Although there are some ablation studies, the analysis of the individual contributions of each component is insufficient, making it difficult to determine the specific impact of each component on the overall performance improvement.

**Questions:**

（1）Could you provide more details on how the neuron starvation problem was diagnosed? Specifically, what metrics or observations were used to identify this issue in SNNs for graph classification?
（2）The paper mentions the use of learnable initial thresholds for neurons. Could you elaborate on how these initial values are set and what specific strategies or heuristics were used to determine them?
（3）Conduct a more thorough ablation study to analyze the independent contributions of each component (e.g., TAG, learnable initial thresholds) to the overall performance. This will help readers understand the significance of each part of the proposed method.
（4）The sensitivity analysis shows variations in performance with different initial thresholds and learning rates. Could you explain why certain thresholds or learning rates were more effective and how they were chosen?
（5）How does TAS-GNN scale with very large graphs in terms of computational efficiency and memory usage? Are there any specific optimizations or techniques used to handle large-scale datasets?
（6）While the paper compares TAS-GNN with several baseline methods, could you consider including comparisons with more recent or advanced GNN models that have shown strong performance in graph classification tasks?
（7）Have you tested TAS-GNN on any real-world applications or datasets beyond the ones mentioned? If so, could you share the results and insights gained from these experiments?

**Limitations:**

(1) While the paper discusses the neuron starvation problem and the sensitivity of initial thresholds, it does not explicitly outline the broader limitations of the proposed TAS-GNN method. It would be beneficial to include a dedicated section that explicitly lists and discusses the limitations of the current work.
(2) The paper does not thoroughly address how TAS-GNN scales with extremely large datasets or very high-dimensional graphs. Including an analysis of computational complexity and memory usage for larger graphs would provide a clearer understanding of the scalability limitations.
(3) While multiple datasets are used, the paper could further discuss the generalizability of TAS-GNN to other types of graph-based tasks beyond classification, such as regression, clustering, or even dynamic graphs.

---

> ### Author Rebuttal · Authors · 2024-08-07
>
> Thank you. We appreciate acknowledging the strength in our work and providing detailed feedbacks. We would like to answer the questions as follows.
>
> ### **W1/Q7/L3.  Extensibility to other datasets and application areas.**
>
> Thank you. We extended the evaluation to more datasets (IMDB-MULTI, and REDDIT-BINARY) and tasks (regression and clustering). TAS-GNN outperforms the SNN-GNN baselines in most cases. See [Table1] in the attachment.
>
>
>
> The importance of the additional datasets (IMDB-Multi, REDDIT-Binary) lies in their connectivity since they lack node features. Our method proves to be more effective on datasets highly related to connectivity. On the new tasks, the graph regression task is similar to the graph classification, where TAS-GNN outperforms ANN. This demonstrates that resolving the neuron starvation problem can be beneficial for overall graph-level tasks. However, on clustering tasks, the binary information loss results in a performance decrease. Despite this, TAS-GNN still outperforms all SNN baselines.
>
>
> ### **W2/Q2. Description of the training process, especially for updating the initial threshold**
>
> We apologize for the confusion. We believe the original description given in Section 4.3 was confusing especially with ambiguous use of the term “initial threshold”. Here’s our second take with clarification and added details:
>
> 1. During the inference with TAG (Section 4.2), the threshold ($V^g_{th}(t)$) assigned to group $g$ adaptively changes every step by Eq.10. At the inference step 0, the step-zero threshold values $V^g_{th}(0)$ are all initialized to $V_{init}$, a hyperparameter that we referred to as *initial threshold* in the paper.
>
> 2. With the threshold learning scheme in Section 4.3, the $V^g_{th}(0)$ is now a trainable parameter that is initialized at training epoch 0 with $V_{init}$ (instead of the inference step 0). The values $V^g_{th}(0)$ are learned through gradient descent during the training epochs.
>
> The ambiguity arises from two types of initial points – the initial (inference) step and the initial (training) epoch. We will distinguish them as step-zero threshold and epoch-zero threshold. Please also see Algorithm1 in the Appendix A.5.
>
>
> ### **W3/Q3. Additional ablation study for performance**
>
> Thank you. [Table3] in the attachment shows a more detailed ablation study including sections 4.2 (TAG) and 4.3 (threshold learning).
>
> As observed, the “Baseline+TAG” and “Baseline+threshold learning” both outperform “Baseline” in most cases. Interestingly, adding threshold learning alone to the baseline does not improve performance for any model with IMDB-BINARY dataset. We believe this emphasizes our key claim that adequate grouping is important for SNN-GNN performance. Please also see Table 5 of the Appendix A.6 for the impact of gradually moving from “Baseline+threshold learning” (1 degree group) to TAS-GNN (max groups).
>
>
> ### **Q1. Details about diagnosing neuron starvation problem**
>
> Thank you. For each node, there are 128 feature neurons. When the average spike occurrence of those 128 neurons over five timesteps are less than 10%, we diagnosed the node as under starvation. We will clarify them in the paper. When we evaluated IMDB-BINARY dataset with this metric, 93.3% of the neurons suffered from starvation.
>
> ### **Q4. Sensitivity with initial threshold and learning rates**
> We would like to clarify that the TAG method (red lines of Figure 4) is sensitive to the $V_{init}$, but further applying the threshold learning (blue lines of Figure 4) makes it stable. The stableness comes from the fact that the TAG method uses $V_{init}$ as the step-zero threshold, while the addition of threshold learning method uses $V_{init}$ as the epoch-zero threshold. During the training, the step-zero threshold is learned to find a suitable value. Please also see our clarification in W2/Q2.
>
> Regarding the learning rate, a range of 0.01 to 0.05 is effective due to the use of large dropout values, which was needed to reduce oversmoothing. This makes the training favor relatively large learning rates. However, when we use too large learning rates, the overall training process becomes unstable. Please also see [Table4] in the attachment.
>
> ### **Q5/L2. Optimization techniques that should be considered for large scale graphs**
>
> For larger graphs, the additional memory cost of #(unique degrees) X #(feature dim) is needed per layer to store the per-group thresholds. When the #(unique degrees) grows too large, merging vertices of similar degrees into groups would reduce the cost. Please also see Table 5 in the Appendix A.6 for the sensitivity to the #groups.
>
> ### **Q6. Performance evaluation on more recent or advanced GNN models**
>
> Thank you. We added experimental results with advanced models using DeepGCN [1] and UGformerV2 [2] in [Table2] in the attachment, where TAS-GNN maintains performance benefits.
>
>
> DeepGCN is a representative architecture that uses residual connections to address the over-smoothing problem. UGformer is a graph-transformer architecture that applies a self-attention mechanism with GNN layers. We replaced all GNN layers in these models with corresponding SNN-GNN layers.
>
>
> [1] Li, Guohao, et al. "Deepgcns: Can gcns go as deep as cnns?." ICCV. 2019.
>
> [2] Nguyen et al. "Universal graph transformer self-attention networks." WWW. 2022.
>
> ### **L1.  Additional limitations of TAS-GNN**
>
> Thank you. Our limitations could be itemized as below.
> * This work mainly focuses on graph classification tasks. We believe the proposed TAS-GNN can be extended to other tasks, such as graph regression. We performed additional experiments on them, and will add them in the paper.
> * The performance evaluation is conducted on relatively small graphs. The proposed method has the potential to be applied to extremely large graphs, we leave them as out future work.
> * The proposed method increases GNN training time due to additional learnable parameters. However, we the overhead is negligible.

---

> > ### Comment · Reviewer_icJz · 2024-08-11
> > **TAS-GNN: Topology-Aware Spiking Graph Neural Networks for Graph Classification**
> >
> > Thank you for the response. I have read the response as well as the reviews and rebuttals of other reviewers. I will stand by my original recommendation.

---

> > > ### Author Response · Authors · 2024-08-11
> > >
> > > Thank you for taking the time to share your thoughts. We genuinely appreciate your comments and feedback, and we're confident your insights will help us further enhance our work.

---

### Official Review · Reviewer_4pSD · 2024-07-11

**Soundness:** 2
**Presentation:** 2
**Contribution:** 2
**Rating:** 3
**Confidence:** 4

**Summary:**

The paper presents a novel approach called TAS-GNN (Topology-Aware Spiking Graph Neural Networks) to address the performance gap between spiking neural networks (SNNs) and artificial neural networks (ANNs) in graph classification tasks. The authors identify a "starvation" problem in spiking neurons within GNNs, where many neurons do not emit any spikes during inference, leading to severe information loss. This problem is more pronounced in graph classification tasks, where the test set graphs are independent from the training set, unlike in transductive or inductive learning settings.

**Strengths:**

1.	This paper identifies a critical "starvation" problem in spiking neurons within Graph Neural Networks (GNNs), where many neurons do not emit any spikes during inference, leading to severe information loss. This problem is more pronounced in graph classification tasks, where the test set graphs are independent from the training set
2.	The paper proposes a novel approach called TAS-GNN (Topology-Aware Spiking Graph Neural Networks) to address the graph classification problem.

**Weaknesses:**

1.	The authors use the node degree instead of the concept of topology, there’s a large gap between the graph topology and node degree.
2.	The authors solve the graph classification task as a contribution, which is not a significant challenge for spiking graph neural networks.
3.	The advantage of Spiking Neural Networks (SNN) is their low energy consumption. However, the paper does not mention the feature, so it is unclear why graph neural networks should be combined with SNN. The motivation behind TAS-GNN is not clear.

**Questions:**

The important points listed in weakness 1-3.

**Limitations:**

The authors adequately addressed the limitations.  The authors should discuss more details of the potential negative societal impact of the work.

---

> ### Author Rebuttal · Authors · 2024-08-07
>
> We thank the reviewer for the acknowledging our novelty of the work and providing constructive feedbacks. We have addressed the comments as below. We will revise our paper according to the rebuttal.
>
> ### **W1. Gap between graph topology and node degree**
>
> We used node degree information for one of the representative graph topology properties. As the reviewer mentioned, degree information is not the entirety of topology information. Thus, we will refine our claim to reflect that we used degree information instead of topology. The reason why we initially used the term topology is that we regarded degree information as a representative feature of graph topology and thought topology would be better understood than degree to readers. For instance, several papers [1, 2, 3, 4] have utilized degree as a core information representing the topology.
>
> [1] Bounova, Gergana, et al. "Overview of Metrics and Their Correlation Patterns for Multiple-Metric Topology Analysis on Heterogeneous Graph Ensembles." Physical Review E, 2012,
>
> [2] Tangmunarunkit, Hongsuda, et al. "Network Topology Generators: Degree-Based vs. Structural." SIGCOMM Computer Communication Review, 2002
>
> [3] Zhang, X., et al. "On Degree Based Topological Properties of Two Carbon Nanotubes." Polycyclic Aromatic Compounds, 2020.
>
> [4] Zhou, Shi, and Raúl J. Mondragón. "Accurately Modeling the Internet Topology." Physical Review E, 2004.
>
>
>
> ### **W2. Significance of challenge for graph-classification task**
> In this work, the main challenge is to achieve high performance (i.e., accuracy) in the graph classification task using SNN. This is different from simply adopting existing SNN-GNN to build a working example on graph classification task, which does not pose a significant challenge. Rather, the challenge was that simple adoption of previous works (e.g., SpikingGNN, SpikeNet, PGNN) causes severe accuracy degradation. Thus, our contribution would not be supporting the task itself but identifying the neuron starvation problem and proposing techniques to address it.
>
>
>
> ### **W3. Consideration of energy efficiency**
>
> Thanks for the great idea. We compared the energy consumption in [Table5] below. The TAS-GNN model shows significant energy efficiency (69%-99%) compared to ANN architectures.
>
> ||||||||
> |---|---|---|---|---|---|---|
> |**ENERGY**| (**mJ**)| **MUTAG**|**PROTEINS**|**ENZYMES**|**NCI1**|**IMDB-BINARY**|
> | **GCN**| **ANN**| 0.53 |6.92 |3.29|21.41|3.16 |
> ||**TAS-GNN**|0.10|0.94 |0.52 |5.28 |0.70 |
> ||**Reduction**|**82.17%**|**86.36%**|**84.29%**|**75.35%**|**77.72%**|
> | **GAT**|**ANN**|0.33 |4.59|2.42|15.55|2.20|
> ||**TAS-GNN**|0.07|0.05|0.34|4.75|0.55|
> ||**Reduction**|**79.96%**|**98.82%**|**85.83%**|**69.44%** |**74.89%** |
> |**GIN**|**ANN**|0.39|4.96|2.33|15.26|2.24|
> ||**TAS-GNN**|0.05|0.02 |0.14|1.67|0.06|
> ||**Reduction**| **87.14%**|**99.64%**|**94.14%**|**89.04%**|**97.48%** |
>
> [Table5] Energy consumption table
>
> We observe significant energy reduction in the PROTEINS dataset with GIN architectures, showing a 99.64% reduction. In contrast, we observe that our worst case for energy consumption showed in the NCI1 dataset with GAT architecture, indicating 69.44% energy reduction. Since the GAT architecture requires more information to learn its attention mechanisms, the spike frequency was higher than in other architectures. Additionally, we found that NCI1 results in particularly frequent spikes among the datasets, which led to less energy reduction.
>
> The theoretical estimations we provided are based on [5,6], which are widely used for SNN energy consumption analysis. We calculated each layer's sparsity $\gamma$ and FLOPs (floating point operations). Assuming MAC and AC operations are implemented on 45nm hardware, we handled $E_{MAC}$ = 4.6pJ, $E_{AC}$=0.9pJ. The energy consumption of SNN is calculated with $E_{AC} \times \gamma \times \text{FLOPs}$. As spike sparsity in our experiment varied greatly depending on GNN architectures and datasets, we evaluated each spike sparsity.
>
> [5] Horowitz, M. “1.1 Computing’s Energy Problem (and What We Can Do About It).” 2014 IEEE International Conference on Solid-State Circuits Conference, 2014
>
> [6] Yao, M et al. “Attention Spiking Neural Networks.” IEEE Transactions on Pattern Analysis and Machine Intelligence, 2023
>
> ### **L1. Negative societal impact of the work**
>
> Thanks for the suggestion. We believe the negative societal impact could be discussed in the following ways.
> * Our research uses social network graphs like REDDIT-BINARY, and this could potentially be misused to screen for ideologies and biases, leading to negative effects such as infringing on human freedom.
> * Despite our efforts to focus on energy reduction, our research could contribute to environmental problems due to carbon dioxide emissions during the training process.
>
>
> However, these problems are not unique to our work; it is an issue faced by all graph neural networks (GNNs). These GNNs can unintentionally retain biases present in the data they learn from, resulting in ethical and societal concerns that need to be addressed by everyone working in this field.

---

> > ### Comment · Reviewer_4pSD · 2024-08-10
> > **Response to Author Rebuttal**
> >
> > The authors give a detailed reply to weaknesses 1-3. Since these three issues are crucial to the motivation and novelty of the paper, judging from the rebuttal statement, the author needs to make a large amount of modifications to address these issues. The current version is not suitable for public publication.

---

> > > ### Author Response · Authors · 2024-08-10
> > > **Response to Reviewer 4pSD**
> > >
> > > Thank you for reading our rebuttal and sharing your thoughts. However, could we ask for a reconsideration? We believe the amount of revisions necessary from the current version would not be very extensive.
> > >
> > > Firstly, regarding W1, it would be sufficient to replace our term topology with a degree, as we used topology to refer to degree information in our explanation within the paper.
> > >
> > > Secondly, regarding W2, we believe the significance of the graph classification task is already clearly demonstrated by the accuracy degradation of other baselines. Therefore, we believe there is less need for a major revision.
> > >
> > > Finally, regarding W3, we think this concern can be addressed by simply adding this experiment table to the main manuscript. Adding these experiments to the paper will certainly be valuable. However, they would not significantly affect the overall flow of our paper, because reducing energy consumption is already a fundamental advantage of SNNs, and we focus on improving their accuracy for practicality.

---

### Official Review · Reviewer_MTu6 · 2024-07-11

**Soundness:** 3
**Presentation:** 3
**Contribution:** 3
**Rating:** 6
**Confidence:** 4

**Summary:**

This paper proposes topology-aware spiking graph neural networks with adaptive thresholds based on a group of neurons for graph classification. The paper first diagnoses the poor performance as the existence of neurons under starvation caused by the graph structure. Then the paper proposes the adaptive threshold among neurons partitioned by degrees, as well as the learnable initial threshold and decay rate to reduce the sensitivity. Experiments on several datasets show superior performance of the proposed method.

**Strengths:**

1. This paper proposes the first SNN design to target graph classification.

2. This paper identifies the starvation problem and proposes a novel topology-aware group-adaptive technique.

3. Experiments show superior performance on several datasets, some outperforming ANNs.

**Weaknesses:**

1. The proposed method seems to be a hybrid ANN-SNN model rather than a pure SNN design. The paper did not discuss how this will affect the deployment of the model on potential neuromorphic hardware, since SNNs mainly target those hardware to obtain energy efficiency.

2. The paper did not discuss the (theoretical) energy efficiency estimation, which is a major motivation for considering SNNs as stated in Introduction.

3. Or if the motivation is to get models with better performance than ANN, then Table 1 does not include state-of-the-art ANN results for comparisons.

**Questions:**

Some recent works also study SNN for link prediction tasks in graphs [1] besides node-level classification, which may be discussed.

[1] Temporal Spiking Neural Networks with Synaptic Delay for Graph Reasoning. ICML 2024.

**Limitations:**

The authors discussed limitations in Appendix A.1.

---

> ### Author Rebuttal · Authors · 2024-08-07
>
> We thank the reviewer for acknowledging our contributions and positive feedback. We faithfully address the comments below.
>
> ### **W1: The proposed method seems to be a hybrid ANN-SNN model rather than a pure SNN design.**
>
>
> We proposed TAS-GNN as a pure SNN design, which shares almost the same backbone architecture with the existing SNN-GNN family (SpikingGCN [1], SpikeNet [2], Spiking GATs [3]). We apologize for such confusion, where we suspect the below two reasons.
>
> **1. Figure 2 shows GNN layer + SNN layer separately.**
> Although Figure 2 separately depicts a GNN layer and SNN layer, the ‘GNN layer’ was not intended to indicate the use of ANN, but just to show a GNN-style message passing occurs here. The term ‘SNN layer’ was merely used to indicate that membrane exists there. All layers operate and communicate by spikes and no ANN layer is involved. We will clarify this and fix the figure accordingly.
>
> **2. An extra operation of aggregation and combination in the last layer.**
> We wanted to note that it comprises a different structure from intermediate layers, not to indicate the use of ANN layers. For fair comparisons, we followed the convention of many SNN architectures that allow a few additional operations after the ultimate-layer neurons [4,5,6]. However, our architecture is orthogonal to such configuration and the same advantage remains over the baselines without those.
>
> If the reviewer had other reasons for considering our model a hybrid, please let us know so that we can clarify.
>
> [1] Zhu, Zulun, et al. “Spiking Graph Convolutional Networks.” IJCAI, 2022.
>
> [2] Li, Jintang, et al. “Scaling Up Dynamic Graph Representation Learning via Spiking Neural Networks.” AAAI, 2023.
>
> [3] Wang, Beibei, and Bo Jiang. “Spiking GATs: Learning Graph Attentions via Spiking Neural Network.” arXiv:2209.13539, 2022.
>
> [4] Zhou, Zhaokun, et al. “Spikformer: When Spiking Neural Network Meets Transformer.” ICLR, 2023.
>
> [5] Zhou, Zhaokun, et al. “Spikformer v2: Join the High Accuracy Club on ImageNet with an SNN Ticket.” arXiv:2401.02020, 2024.
>
> [6] Shi, Xinyu, Zecheng Hao, and Zhaofei Yu. “SpikingResformer: Bridging ResNet and Vision Transformer in Spiking Neural Networks.” CVPR, 2024.
>
>
>
> ### **W2. Is the motivation for the energy efficiency of SNN? If so, show comparison over ANNs.**
>
> Thanks for the great idea. We compared the energy consumption in [Table5] below. The TAS-GNN model shows significant energy efficiency (69%-99%) compared to ANN architectures.
>
> ||||||||
> |---|---|---|---|---|---|---|
> |**ENERGY**| (**mJ**)| **MUTAG**|**PROTEINS**|**ENZYMES**|**NCI1**|**IMDB-BINARY**|
> | **GCN**| **ANN**| 0.53 |6.92 |3.29|21.41|3.16 |
> ||**TAS-GNN**|0.10|0.94 |0.52 |5.28 |0.70 |
> ||**Reduction**|**82.17%**|**86.36%**|**84.29%**|**75.35%**|**77.72%**|
> | **GAT**|**ANN**|0.33 |4.59|2.42|15.55|2.20|
> ||**TAS-GNN**|0.07|0.05|0.34|4.75|0.55|
> ||**Reduction**|**79.96%**|**98.82%**|**85.83%**|**69.44%** |**74.89%** |
> |**GIN**|**ANN**|0.39|4.96|2.33|15.26|2.24|
> ||**TAS-GNN**|0.05|0.02 |0.14|1.67|0.06|
> ||**Reduction**| **87.14%**|**99.64%**|**94.14%**|**89.04%**|**97.48%** |
>
> [Table5] Energy consumption table
>
> We observe significant energy reduction in the PROTEINS dataset with GIN architectures, showing a 99.64% reduction. In contrast, we observe that our worst case for energy consumption showed in the NCI1 dataset with GAT architecture, indicating 69.44% energy reduction. Since the GAT architecture requires more information to learn its attention mechanisms, the spike frequency was higher than in other architectures. Additionally, we found that NCI1 results in particularly frequent spikes among the datasets, which led to less energy reduction.
>
> The theoretical estimations we provided are based on [7,8], which are widely used for SNN energy consumption analysis. We calculated each layer's sparsity $\gamma$ and FLOPs (floating point operations). Assuming MAC and AC operations are implemented on 45nm hardware, we handled $E_{MAC}$ = 4.6pJ, $E_{AC}$=0.9pJ. The energy consumption of SNN is calculated with $E_{AC} \times \gamma \times \text{FLOPs}$. As spike sparsity in our experiment varied greatly depending on GNN architectures and datasets, we evaluated each spike sparsity.
>
> [7] Horowitz, M. “1.1 Computing’s Energy Problem (and What We Can Do About It).” 2014 IEEE International Conference on Solid-State Circuits Conference, 2014
>
> [8] Yao, M et al. “Attention Spiking Neural Networks.” IEEE Transactions on Pattern Analysis and Machine Intelligence, 2023
>
> ### **W3. Is the motivation to get better performance over ANN algorithms?**
> Our motivation is not to outperform ANNs, but to build a SNN-GNN architecture that can outperform existing baselines and reduce the existing gap to ANN performance. As the reviewer has correctly assumed in W2, the advantage of ANN lies in energy efficiency. However, we are excited to find that several datasets such as IMDB-BINARY results show superior performance compared to ANN under the same backbone architecture. Our performance is also comparable with the leaderboard results using ANNs (https://paperswithcode.com/sota/graph-classification-on-imdb-b.)
>
> ### **Q1. Discussion on recent study**
>
> Thanks for the great suggestion to discuss papers that support the SNN-GNN area [9]. We will add the discussion below.
>
> "While the node classification task is the most commonly addressed, a recent work GRSNN [9] explores the link prediction task to achieve energy efficiency using SNNs in knowledge graphs, demonstrating that incorporating synaptic delays into SNNs allows for effective relational information processing with significant energy savings."
>
> [9] Xiao, Mingqing, et al. “Temporal Spiking Neural Networks with Synaptic Delay for Graph Reasoning.” ICML, 2024.

---

> > ### Comment · Reviewer_MTu6 · 2024-08-12
> >
> > I would like to thank the authors for their detailed responses, clarification, and additional results. Most of my questions are solved and I raise my score.

---

> > > ### Author Response · Authors · 2024-08-13
> > >
> > > Thank you! We greatly appreciate the reviewer's thorough consideration of our response and the valuable insights shared with us.

---

### Author Rebuttal · Authors · 2024-08-07

We sincerely thank all the reviewers for dedicating their time to evaluate our work. We are encouraged that they found our approach to be novel in developing TAS-GNN (MTu6, 4pSD, icJz, 5K64), with clear motivation demonstrated by diagnosing neuron starvation (4pSD, icJz) and competitive performance compared to other baselines (MTu6, icJz, 5K64). Our rebuttal can be summarized as follows:

* Theoretical energy consumption comparison between TAS-GNN and ANN architectures
* Clarification on the usage of the term ‘topology’
* Analysis of the performance across different datasets and tasks
* Experiments with additional GNN architectures
* Additional ablation studies for performance

Please note that the attached PDF contains additional experimental results, which we explain in detail in each response.
For the remaining rebuttal period, we will try our best to answer any further questions and discussions on the topic.

---

### Decision · Program_Chairs · 2024-09-25

**Decision:**

Reject

**Comment:**

The paper proposes Topology-Aware Spiking Graph Neural Networks(TAS-GNN) to alleviate the neurons under starvation problem. The problem is very severe in the graph classification tasks. TAS-GNN leverages the topology of the graph to improve the performance of spiking neural networks in graph classification tasks. Experiments on diverse datasets show the empirical performance of the TAS-GNN.

During the review stage, several concerns are proposed by reviewers which are partially addressed by the authors with a lot of necessary changes for the current submission. After carefully considering all evaluations of other paper in the same batch, I recommend rejection for this submission. I hope the authors can improve the paper and submit it to a future venue.